# Reconciling Nature-Technology-Child Connections: Smart Cities and the Necessity of a New Paradigm of Nature-Sensitive Technologies for Today's Children

Raisa Sultana [1,*] and Scott Hawken [2]

1   Department of Geography and Environment, Faculty of Earth and Environmental Sciences,
    University of Dhaka, Dhaka 1000, Bangladesh
2   Landscape Architecture Program, School of Architecture and Civil Engineering, University of Adelaide,
    Adelaide, SA 5005, Australia; scott.hawken@adelaide.edu.au
*   Correspondence: raisasultana@du.ac.bd

**Abstract:** There is a serious and problematic disconnection between children and the natural environment. This has been documented across various disciplines and fields of endeavour, including science, the creative arts, the social sciences, education, design, and the humanities. The nature–people disconnection is particularly concerning at this present juncture when understanding and advocating for the natural environment is necessary to address global environmental crises. Smart cities have, to date, focused on business and economic directions. In recent times, there has been an emerging awareness that such technologically advanced urban environments must link to and inspire an understanding and care for nature in more profound and meaningful ways. Therefore, this paper aims to identify opportunities and discuss how technology can improve this interaction through advancing and implementing nature-positive and nature-sensitive technologies through a critical review of the literature spanning smart cities, children, and nature-based technologies. Such linkages can serve as a driving force behind the transformation of cities as they adapt to support initiatives, such as the post-2020 biodiversity agenda.

**Keywords:** child-nature relationship; nature disconnectedness; nature-sensitive technologies; smart cities; nature sensitive design; IoT wearables; mobile devices; augmented reality; gamified activities; landscape technologies

## 1. Introduction

Interactions with urban natural environments are beneficial for children's health and well-being and create place attachment and appreciation for the conservation of the natural environment [1–3]. Exposure to nature positively influences children's physical and mental health, including self-esteem, emotional well-being, and quality of life, happiness, and attention deficit disorders [4–6].

Although many types of research show that children benefit from nature exposure, today's children spend less time in natural environments than ever before. In developed countries, children in the current and future generations are separated from the environment through both environmental and behavioural factors [5–8]. The terms "nature deficit disorder" and "ecophobia" [8] have been invented to describe nature deprivation [7]. Some experts believe that children can benefit from spending less time on technology and more time outside because this can help to connect them with nature [9,10].

In this paper, nature is defined and interpreted as any outdoor area in an urban setting, including natural elements, such as plants, grass, trees or biodiversity that children can utilize for recreation, play, creative activities and educational purposes. These natural settings can include backyards, neighbourhood parks, local reserves, small and large central parks, school playgrounds, or forest areas. Such urban spaces can benefit from careful

design to enhance opportunities for play and recreational activity and to simultaneously nurture an appreciation of biodiversity and nature. This holistic approach, integrating consideration of children, nature and biodiversity, can facilitate education and nature-positive values [11]. Whereas nature-based technology is regarded as a digital application tool that can help connect children to the natural environment [5,11–17]. These can include both tangible (e.g., smartphones, tablets, software, and tracking devices) and intangible tools, such as software, applications, or augmented reality applications.

The implementation of intelligent technologies is the prerequisite for smart city development and has enabled the transformation of cities around the world. Urban governments aim to implement smart city policies and projects for a number of reasons, including the support of businesses, communities, and citizens [18]. Leading smart cities combine sustainable development fundamentals with information and communication technology (ICT), cloud computing, big data, internet of things (IoT), and technology [19,20] in six areas: economy, governance, people, transportation, environment, and living [20–22]. To be effective smart cities require careful consideration of social landscapes, human well-being, and the built environment for sustainable living [20,23]. In recent times there is an emerging awareness that such technologically advanced urban environments must link to and inspire an understanding and care for nature in more profound and meaningful ways. A key question, in this case is "how can technology enhance children's nature explorations, discoveries, and connections given the already challenging environment for young children and the increasing focus on technologies in early years?" [24]. Research suggests that we should examine the repercussions such a paradigm shift could have on children's lives in the long run: including their value of natural space, the quality and frequency of their interactions and relationships with nature, and their enthusiasm to engage in nature-based activities [5,9–17].

Smart cities have, to date, focused on business and economic directions rather than supporting key demographics, such as children, and this has been noted in the literature [25]. For example, Han et al. [26] suggested that smart city agendas can "reduce cities to a one-dimensional business model and series of metrics", whilst Söderström et al. [27] argue that companies, such as IBM, have co-opted the smart cities agenda as a form of corporate storytelling by presenting technological innovation as a pragmatic and uncontroversial advance in urban development and management [27]. In his 2014 paper, Kitchin [28] elaborates on the shortcomings of much of the literature and the urban scholarship of the smart city [27–29], suggesting that we need to address the needs and benefits for all citizens, including children [14,27,29].

Addressing the alarming consequence of the disconnect with nature, thus, requires an appreciation of how emerging technologies can motivate children to experience significant and regular interactions with urban natural environments. The paper aims to give insight into the extent to which technologies can help children connect with natural environments more often, more deeply and more safely. Using a systematic and critical review of the literature, we have identified relevant nature-sensitive technology applications. Following this, we then discussed these technologies in terms of their characteristics and effectiveness in enhancing and inspiring children's connection to nature.

## 2. Materials and Methods

### 2.1. Search Strategy

To answer the question, "how might smart technologies enhance nature children connections?" we reviewed the current literature both systematically and critically. The search process was informed across the themes of nature, children, and technology in smart cities and sought to find and assess the literature that critically presented linkages between these three themes. Keywords and phrases, such as "nature-based technology" and other related terms, are shown in Table 1. The authors used these keywords to identify a set of 62 papers from the relevant published literature collated from four high-impact electronic databases—ScienceDirect, Web of Science, IEEE, and Scopus, between 2003 and 2022.



**Table 1.** Keywords search queries.

|  | Themes | Keywords |
|---|---|---|
| MAIN | "Children in Smart cities" | 'Smart city children' OR<br>'Smart technologies' OR<br>'Digital Placemaking' OR<br>'Children recreation in smart cities' OR<br>'Nature connectedness in Smart city' OR<br>'Smart city concept' OR<br>'Children and Technology' OR<br>'Connecting children with nature' |
| AND | "Nature-Based Technologies" | 'Nature-based play' OR<br>'Human-nature relationship' OR<br>'ICT' OR<br>'Information Technology' OR<br>'Nature-based solutions' OR<br>'Technology in Nature Education' OR<br>'Mobile Technology' OR<br>'Augmented Reality' OR<br>'Wilderness Technology' OR<br>'Ecofriendly Technology' |

*2.2. Inclusion and Exclusion Characteristics*

Inclusion and exclusion characteristics were developed to ensure that reviewed papers were relevant. The keywords used are listed in Table 2. The selected papers were peer-reviewed articles published in the English language, translated versions were also accepted where available. Articles between the years 2003 to 2022 were selected to ensure they were current and relevant. Age range was a key inclusion criteria (1–18 years) and based on the definition of children given by UNICEF [30]. Papers whose sample contained population ranges not within this range were excluded.

**Table 2.** Inclusion and Exclusion Factors.

| Characteristics for Inclusion | Characteristics for Exclusion |
|---|---|
| • English language articles<br>• Peer-reviewed journal articles and conference proceedings<br>• Full text available on electronic media<br>• Available within the selected databases<br>• Articles with both research and review topics<br>• Articles that were published between 2003 and 2022<br>• Articles that included nature-based technology that can be used and perceived by children only. | • Not available in English<br>• Duplicate articles between various databases<br>• Research objectives of the papers are not clearly mentioned<br>• Age ranges considered are outside of 1–18 years (i.e., not children)<br>• Unrelated and irrelevant topics and themes. |

The technologies identified in this paper as 'nature-connected technologies' were considered for their relevance to children. Technologies were also selected or excluded based on their current or emerging usage. For instance, outdated or obsolete technologies were excluded. Technologies that are only used by adults or are not related to nature were excluded from this study. Other supporting information was gathered from several conference proceedings to support the review process.

Other supporting information was gathered from different websites or applications developed by individual commercial organizations and blogs which contained information about nature-based applications. Around 30 nature-based applications were found in usage, and the authors included and cited nine websites. Hence, the paper aims to review those technologies that could help children better and more effectively connect with their environment. In this way, a total of 62 publications were obtained.

### 2.3. Data Extraction and Analysis

The published articles that successfully matched the eligibility characteristics went through a data extraction and analysis process, which was completed by the authors. Data extraction is crucial and needs to be relevant to the realm of the research topic. The data extraction procedure was conducted based on the criteria shown in Table 3.

**Table 3.** Data analysis themes summary table.

| Themes/Categories | Number of Papers | Percentage (%) | Cited Authors |
|---|---|---|---|
| **Environmental Awareness and Conservation** | 7 | 11 | [1,4,31–35] |
| **Nature Education and learning** | 22 | 35 | [9,13,15,16,24,36–51] |
| **Nature Play** | 18 | 29 | [5,11,12,17,52–65] |
| **Collaboration** | 3 | 5 | [66–68] |
| **Nature Exploration** | 12 | 19 | [2,3,19,69–77] |

The retrieval and identification of the peer-reviewed papers were followed by data extraction and analysis. This was accomplished by doing a qualitative categorization of the various papers and creating a concise and descriptive summary of key aspects of the reviewed publications. In order to make recommendations and contribute to the body of knowledge, these data were categorized and analyzed in accordance with the efficacy and limitations of nature-based technology for children. The literature was classified into five categories of nature-child relationships. These are Environmental Awareness and Conservation, Nature Education and Learning, Nature Play, Collaboration, and Nature Exploration. These themes were further reviewed in the discussion in terms of their efficacy, limitations, and benefits for enhancing child nature relationships.

## 3. Results

### 3.1. Three Types of Nature Experiences and Technology

Within cities, natural spaces have been neglected and in this process, become less attractive and inaccessible for urban citizens, resulting in the disappearance of nature play and a narrowing of environmental sensitivity and awareness [66,67,78]. In this paper, we argue that smart digital technologies can play a significant role in recognizing the challenges in child–nature connection opportunities that children are facing in the present day. Such arguments run counter to common perceptions that smart digital technology is a hindrance or barrier to children's interaction in nature. As people increasingly rely on digital networks in their everyday lives, a new layer of the urban environment has emerged—a digital environment that is mediated through various technologies. This has led to a change in how the public perceives, thinks about, and interacts with nature [28]. In the literature various researchers have sought to identify how nature connectedness can be achieved via technological interventions. Based on the literature from the reviewed articles, the authors of this paper identified smart technologies which can be applied in three arenas: nature education, nature play and recreational activity, and natural exploration through personal discoveries.

#### 3.1.1. Nature Education and Smart Technology

Modern technology can assist with nature education by making nature less threatening and more comfortable whilst expanding accessibility and helping draw attention to different details and diverse dimensions of nature [52,68]. A select range of studies demonstrates that when used thoughtfully, technology can help build and enhance nature–children's relationships. With devices, such as tablets and smartphones, children can become involved and guided to take photographs and send GPS coordinates to the teachers or educators [13]. Stewart et al. [52] showed that enabling the opportunity to use iPads in natural spaces motivated kindergarten children to closely explore the natural spaces

around them, concentrate on specific aspects of the natural world, and capture photographs to inculcate classroom discussions and learning, which led to significant opportunities for environmental education and nature connectedness. Nature and outdoor based education has also been shown to reduce stress in comparison to indoor based learning [52].

Technology can help connect children and adults through digitally mediated practices, such as podcasting, digital recording, and gamification. This can help them learn more about the world around them and overcome barriers, such as geographic, cultural, and emotional ones [36]. By comparing two teaching methods—a traditional pen-and-paper method, and another using mobile technology; some researchers found that mobile technology helped to engage students more, increased their excitement about learning, and helped them understand environmental issues better [15,52]. Stewart and Maguire 2020 carried out a three-year field experiment with kindergarten children that involved them completing creative nature activities with tablet applications. The results showed that by using tablets (or smartphones) during nature play, it empowered them to have more collaborative, creative, and meaningful experiences. Applications involved taking photographs, making videos, and recording a wide range of their creative ventures. Later, the pictures of plants, flowers, and natural landscapes were used in a project named "nature book", which was appreciated by the educators of the kindergarten [52].

Learning about nature through technology is supported by the study of Plowman and McPake [16] as well. They suggested that reciprocity with technologies can facilitate various forms of children learning. Johnston and Highfield, 2017 presented a case study where a child explored the natural world through digital technology, such as mobile and photo editing-based applications. For instance the research paper described how one child took a photograph of a blue-coloured, jellyfish on the beach which was an unknown species to him. Later, the child, along with his father, explored this sea creature and its related information, e.g., habitat, and varieties, made a photo collage using a mobile application, and shared it with his peers. This approach encourages increasing self-efficacy and academic achievement and aids social, cognitive, and emotional development [24,37,52]. The Explore! is a mobile-learning system that is operated by students during an archaeological park visit to join an excursion game, assist students in learning more about historical concepts while playing and making visits to archaeological sites more exciting and thrilling [17]. The opportunity to blog about their experiences helped motivate the students to pay closer attention to the world around them, collect and analyse data, and make discoveries. Similarly, The EduPARK project is an excellent example of how smartphones, along with augmented reality, can promote genuine nature learning (i.e., the study of various plant and animal species) by enhancing opportunities to engage with available urban natural spaces [38].

Technologies, which can also be used in natural spaces, include augmented reality tools, such as 'Google Glass'. This technology consist of a pair of glasses with a system that projects information on the inside of the glasses. It transmits images without obstructing the view directly to the eye and allows users to see what surrounds them while superimposing relevant data about those real world elements and spaces. This technology has demonstrated success in enhancing enthusiasm and engagement during school and fieldwork activities. For instance, in one case, students were assigned by their teachers to visit an urban park and make a checklist regarding what they saw in their natural surroundings with the help of Google Glass. Teachers reported no difficulties found with the use of the technology at the elementary school level. Students successfully managed to carry out their assigned tasks [38]. Nonetheless, it is important to note that the use of technology should always be implemented with consideration of the individual needs and preferences of the learners and must enhance, rather than replace, the natural environment and social interactions.

### 3.1.2. Nature Play and Recreational Activity and Smart Technologies

As smart digital technology is an increasingly ubiquitous component of children's lives, examining the potential for these digital tools to support or facilitate nature-play is necessary, particularly in cities. Children have a dynamic relationship with their surround-

ings. Modern digital technologies e can support authentic play-based learning in such natural environments. Some studies have been conducted which showed technology as a positive tool to bolster children's growth, abilities, and strength in play-based and recreational activities [12,19,36]. For example, a new project called "A New Sense of Place?" has helped increase the potential of contemporary mobile technologies to provide opportunities for children's play and mobility [19]. Another study conducted by O'Loughlin et al. [36] analysed how a simple recorded video can influence children's physical education and learning. Children were given the video to provide feedback, assess themselves, and improve their skills after playing. This approach not only motivated the active students but also left a positive impact on the children who are usually reluctant to participate in any physical education-related activity (e.g., bushwalking or swimming).

The possibilities for digital tools to be designed to encourage nature play or support child nature interaction have been a focus for some scholars. For example, numerous mobile applications and tangible tools have been invented and designed to encourage children to engage in social and physical play—a reaction to the increase in children's sedentary forms of indoor recreation. Some experts installed interactive digital play technology in a school playground and discussed how this supported and mediated the active engagement of children with natural spaces [13].

Today's urban parents have less time to accompany their children to local and distant parks and playgrounds; therefore, children can be deprived of autonomous play due to parents' safety concerns. A smart digital intervention that facilitates parents and families to access urban neighbourhood play places' geolocation data has proven to be a useful supporting technology for urban parents. Details may include playing boundaries, child-only zones, designated safe areas, landmarks, parent zones, and the demarcation of areas. As play is an independent and child-directed practice, preserving children's agency should be given prime importance while introducing digital interventions in contemporary design [11,12].

Numerous examples of smart digital technologies invented and designed for urban nature settings frame children's play activities with fixed goals or encouraged activities, as observed in gamification applications [68]. Gamification is a way to encourage participation by adding playful features of games into other otherwise serious contexts. For example, one might use game design elements, such as points scoring, challenges, and competition, to encourage participation in activities, such as eat healthily, exercising, or saving money [68,69]. Likewise gamification can make exploration of nature more interesting for adolescents who are used to spending their time on screens. Games and entertainment can be enjoyed using smartphones devices and digital screens, but with the emergence of place-based games, the real world has become a game board [53,68]. This entertainment-filled approach can encourage children toexperience natural places, whilst inspiring an educational narrative, and giving children a sense of agency over the activity. Gamification can enhance simple activities, such as tracking a checklist of items while at the beach or participating in a city-wide scavenger hunt [68].

When using gamification concepts, it is important to make sure that the focus remains on nature exploration rather than on the virtual game itself [67]. Researchers and programmers are anticipating that the younger generation who are involved in gamified activities will see the value in the thought-provoking and rich sensory experiences that nature can offer [69]. For example, England's National Trust offers a bingo card for children to engage in fun nature activities, including highly sensory events, such as watching the sunset or going for a swim [79].

Some researchers are explore the accessibility for children in different spaces and opportunities for 'independent' play and free movement, particularly in cities, which seem limited indoors only [9,70]. A primary influence on the progression of nature applications was the success of Pokemon Go [54,71], a location-based game that used mobile augmented reality to let players find and catch virtual Pokemon in the real physical world. The game's developers have since suggested a few ways to make the game more environmentally friendly, including adding real species and teaching conservation skills [54].

Within the field of Human–Computer Interaction (HCI), scholars are now exploring how digital intervention and innovations can support children's independent and open play. One example is 'Wobble', an unrestricted play platform invented to support children's (4–6 years old) socio-dramatic play. Several interactive play features and objects were also designed to facilitate children's physical and social play, helping them to explore their creativity and imagination [55,68]. Projects, such as Ambient Wood [38,69] and Savannah, use wireless networks and location-context sensing technology [53]. In Ambient Wood, children can delight in exploring woodland habitats, and this digital activity invites them to explore the physical environment around them in a new way by augmenting it with digital abstractions [56,72].

### 3.1.3. Nature Explorations, Personal Discoveries and Smart Technologies

Although most interactive nature based technologies for children have been planned for formal educational contexts, some recent initiatives have also pursued the development of more unstructured interactions focused on discovery [11]. These tools were designed to help children explore a particular place and were created with a specific didactic goal in mind- something that helps children focus on aspects of the complex natural environment rather than supporting open ended freedom for children to create their own experiences [11]. Mobile Augmented Reality (MAR) systems are very versatile and can be used to explore the environment in many ways. This allows children to learn more effectively by using different senses and modalities, including hands, body postures, eyes, temperature, smell, and audio, which makes learning easier and quicker [39].

Within MAR systems, augmented features can include anything related to the built environment, including both physical and intangible elements. MAR has the potential to provide children with ways to connect more directly and interactively with the natural world [40]. There are several applications that incorporate augmented reality to enhance the participation of students in nature environments [40,73]. Ryokai 2013 studied the ways om which the GreenHat MAR app could assist with the discovery of sustainability issues and biodiversity in nature. They found that, compared to a digital map on the same smartphone, MAR encouraged children (students) to look more meticulously at field sites, and to also have closer interactions with environments and to make personal discoveries within the natural environment [40,73]. For instance, the ARWeather MAR program can create realistic simulations of different weather conditions on different surfaces to support children in learning about weather [80].

Location-based programs, integrating ubiquitous connectivity and augmented reality (AR), offer advanced ways for people to perceive and participate in urban settings [57,78]. The "Immersive Tour Post" technology functions as smart binoculars that allow people to see historical information related to specific sites in Korean cities [78]. By watching and listening to films and recordings of historical events that took place right in front of them, the children were able to learn more about architecture and history in a more personal way. Albrecht et al. [73] have looked at ways to let people stay connected with their environment, by allowing them to hear the sounds around them while adding virtual sounds to enrich their auditory experience [41]. In some cases, modifications need to be made to conventional MAR devices to enable their use within nature. Veas & Kuruijff [74] invented different technical tools to make a handheld MAR device more comfortable, including grip attachments that can be attached to the user's hands and special belts that can be worn to make the device easier to hold even in cold, snowy environments. These examples illustrate how MAR can be used to connect children with diverse environments in a unique way [53,73,74].

With new software technology, different applications have been developed to make use of GPS to navigate to learning resources that are specific to a location. For example, learning resources can include habitats, landmarks, tourist information, and locations for animal sightings [73,81–83]. Applications, such as The Audubon Bird Guide, BirdNet [75], iBird Plus Yard Guide [75,84], eBird by Cornell Lab [75,85], Merlin Bird ID by Cornell

Labs [58,86], etc., can assist children to identify birds by location, colour, shape, habitat, and other characteristics. These technologies let children see the world in a different way by using digital information that's overlaid in their view.

Currently researchers are investigating how technology can help children form attachments to their local natural environments and how interactive digital systems might help them do this. They have developed contemporary participatory smartphone- and tablet-based sensing applications using automated visual identification of botanical species such as Leafsnap, Plantsnap, Picture This, PlantSnap, iNaturalist and Pl@ntNet. These applications facilitate nature exploration by analysing more than 625,000 species of natural plants and cultivated plants [75,76,79,87–89]. Some mobile learning tools allow users to engage with the activity through a variety of senses, such as seeing shapes and colours, feeling textures and forms, and even smelling different species. These tools help motivate children to interact and share their thoughts and feelings about the physical environment.

Some technologies help people learn more about the plants and animals they see while hiking or on other nature adventures [31,77,90]. Activities such as geocaching [19,42,59,91] where players find hidden treasures using global positioning systems (GPS) scan also be utilized to strengthen children's perception of nature [43]. Within these technologies specific nature modules can helpchildren connect with their natural surroundings.

Nature-based mobile applications (apps) that enable children to blog their observations of the natural world are gaining in popularity as recreational participation and informal learning tools, such as iNaturalist [92], Nature Play, and Nature Passport co-developed by IslandWood [19]. Likewise, Savannah is a game that help's children learn about an African savannah and then behave as if they were living in that environment. The game helps them to understand the environment and its wildlife and to learn how to respect and behave in relation to this environment [19,60]. Another initiative, The Urban Tapestries project, enables people to create and customize their own urban spaces in London. The project is currently designed for adult usage, but children could also use it to create their own fun and to develop unique spaces [93].

Agents of Nature is a place-based mobile application developed by a humanitarian organization that seeks to promote children's engagement with nature. By providing them with an immersive experience, the organization hopes to help children develop a stronger connection with the environment and to identifywith the place [32]. In addition, the creators aim to make parks in the United States and Canada more accessible to children through the app. It does this primarily through including questions about plants and animals that are specific to different neighbourhood parks. For example, it might ask the user questions about the shape of the leaves of a particular plant or what kind of animal you might see there. The questions were tested before the study and demonstrated a good degree of success. Therefore, these new technologies can help children get more out of their environment by encouraging them to use them [15,40].

It is children's innate nature to be curious about the world around them and to use their scientific inquiry skills to learn more about things, such as natural systems, processes, and environments. Weather apps, such as Weather Underground, The Weather Channel, Carrot Weather, and Dark sky weather, are powerful mobile applications that can help children learn about weather forecasts, interactive radar, and rain alerts. In addition All Trails: Hike Bike & Run, TrailLink: Bike, Run & Walk, Gaia GPS: Hiking, Hunting Maps [94,95] are some applications that are used to explore the best hiking, camping, running and biking trails, and other nature exploration around the world.

### 3.2. Emergent Technological Types and Natural Settings

In this section, we consider different types of technologies and their potentially use by children in natural settings. These are:

- Augmented reality which has the potential to provide children with ways to connect more directly and interactively with the natural world.

- Multimodality and extended sensors which can detect motion, temperature, or sound and can make natural space more interactive, effective, and fun for children. For instance the Science Spots AR platform offers a possible and inclusive approach to multidisciplinary teaching by allowing students to interact with both real and virtual objects to better understand scientific concepts as part of the outdoor education movement. This approach has the added benefit of creating a game-based environment that is both educational and enjoyable, promoting motivation among students [38].
- Nature-based applications enable children to research, collect, and analyse the natural world. Applications, such as iNaturalist [92], Nature Play, Nature Passport [19], Savannah [19,60], and Agents of Nature [32], are designed to encourage children to spend time outside and help in exploration.
- Communication and information technologies that prioritize the sharing of nature experiences. Multimedia resources, such as videos, images, and interactive simulations, provide opportunities to develop children's creativity and communication skills by providing a wide range of information sources.
- Gamified activities that encourage and place value on problem-solving and sensory experiences that nature can offer. Games often provide immediate feedback and rewards, such as points, badges, or prizes. This can motivate children to keep learning and exploring as they strive to achieve a specific goal or earn rewards. In nature-based games, rewards can be tied to learning about specific plants or animals or to motivate the completion of certain tasks, such as picking up litter or planting trees. Wobble, Ambient Wood, Pokemon Go, Camelot, etc., are some of the popular nature-based games with children at present [54–56,71,72,75].
- Mobile technologies can allow children greater freedom in cities to partake in 'independent' play and free movement. Such technologies may have been limited to indoor contexts in the past. Digital wearables, wireless gear and equipment for outdoor activities, such as hiking, camping, or birdwatching, can make outdoor activities more comfortable and enjoyable, ensures safe mobility, and can help children feel more connected to nature. Location-sharing systems allow parents to keep an eye on their children [11,12,33,61,70].

Whilst indoor and sedentary types of digital recreation can potentially contribute to a reduction in interactions between children and nature, digital technologies used in outdoor environmental settings can also enhance and create a new appreciation and sensitivity towards nature. The use of technology in natural settings can help young students to develop skills, such as reasoning and higher levels of problem-solving, due to software and applications that require higher-order thinking [38,62]. Sandbrook, Adams, and Monteferri's article discussed the use of digital games for biodiversity conservation. The authors provided examples of games that have been developed for this purpose, such as "Stop That Mosquito!", which teaches players about the dangers of mosquito-borne diseases, and "Plant Defenders", which teaches players about plant defense mechanisms and the importance of pollinators [96]. Various studies and initiatives have identified ways to enhance children–nature relationships through technology-oriented nature-based action plans among youth [96]. Eckhoff [63] carried out an activity plan which involved problem-solving tasks by encouraging school-going children to explore the natural environment. The activity included instructions such as "write down different wildflowers and insects you saw in the park", and environment-related questions, such as "what do clouds look like?". In addition, the study found that digital media allowed children to document and extend their play experiences in new ways. For example, children used cameras to capture their play experiences and then used the photos to create storybooks or scrapbooks. The study also found that digital media provided children with opportunities to collaborate and share their play experiences with others. For instance, children used iPads to take photos of their play activities and then shared the photos with their friends and families. Thus, portable technological devices were taken outside to enhance the learning process by making it a more joyful, meaningful, and fun activity compared to the traditional approach [31,63].

Nature-based technologies promote a reciprocal relationship where children receive physical and mental benefits via nature contact and also learn to appreciate and value nature. Providing children with open-ended digital tools supports opportunities to make observations of natural events and document them. In such cases, parents and educators can review the information children gather and any conclusions they come to [52].

Researchers have examined how technology stimulates children's social interaction, concentration, motivation, attention, ability to perform teamwork, and persistence in learning [44]. Collaboration in a group enhances socioemotional development, and nature settings establish themselves as an impactful communicative platform. The study conducted by McAvoy et al. [97] shows how children with cognitive disabilities developed social and recreational skills and levels of participation in natural space with the help of modern technologies. Modern digital applications can enhance interactions between students through sharing and supporting learning activities. For example, Plantsnap is an app that uses image recognition technology to identify different plants and flowers. The application includes a "Kid Mode" feature that simplifies the interface and provides fun facts and educational content about the plants [75,90].

Mobile augmented reality (MAR) is an emerging technology that can help children to interact with and comprehend the environment and objects through the provision of additional contextual information and the superimposition of contextual spatial information [40]. Augmented Reality (AR) is a technology that overlays digital information and computer-generated graphics onto the real world. AR can be experienced through a variety of devices, such as smartphones, tablets, and AR headsets, which makes the experience more interactive and enjoyable for children. Such technology can draw attention to different aspects of nature and introduce alternative and supplementary sensory or scientific information. This subtle and integrated synthesis of contextual information is a powerful way to help children learn more effectively as well as have more fun [39,45]. In addition, an experimental study showed how performance and understanding increase via nature augmented reality Folkestad & O'shea [62], did a qualitative investigation of recorded video footage captured during the Augmented Reality (AR) experience that encompassed both indoor and outdoor components. AR technology involves the overlaying of digital information onto physical surroundings. The AR activity in question required students to interact with the San Diego Museum of ART and the Botanical Gardens, both located in Balboa Park, San Diego. Throughout the experience, pairs of students were recorded via video as they navigated the surroundings. Upon analysis, the results revealed that despite encountering a number of technological obstacles, particularly during the outdoor component of the activity, students maintained a relatively high level of engagement and collaboration.

Technological applications can promote the simultaneous development of critical thinking and a greater appreciation of environmental values, sustainability, and ecosystems. Lai et al. [98] onducted a study called Green Map where QR codes were used to discover exciting materials and information. It showed how nature learning effectiveness can certainly increase among students with the help of these multimedia tools [38,44]. Researchers have acknowledged technological media devices, such as computer-assisted programs, interactive whiteboards, and digital cameras, can be effective tools to support innovative teaching-learning methods and increase children's academic outcomes, operational skills, and cognitive development [16,44,46]. Vernadakis et al. [46] investigated the use of computer-assisted instruction (CAI) in a preschool classroom setting. The CAI program was designed to be engaging and interactive with colourful graphics, animations, games, and sound effects. The activities were also designed to be developmentally appropriate for young children with a focus on hands-on learning and exploration. Siskind et al. [44] discussed technology-based learning experiences for young children, such as using virtual reality to explore different environments and cultures and using digital cameras and tablets to document and reflect on outdoor experiences. Such intellectual engagement can also support emotional attachment, which further leads to place attachment, environment stewardship, and a sense of belonging, which are crucial for biodiversity and nature con-

servation. Plowman and McPake [16], also argue that technology can offer many benefits in terms of engagement, motivation, and learning outcomes as children can use digital cameras and microscopes to explore the natural world. The researchers also cautioned that it should be used in moderation and in conjunction with other teaching methods. They emphasized the importance of balancing screen time with other types of learning activities, such as outdoor play and social interaction.

Children can connect theoretical aspects with real-world and tangible aspects, therefore, growing the probability of a better understanding of abstract concepts [11]. Developing this profound relationship with the surrounding natural world would lead children to grow empathy for the species of plants and animals and their habitats. Taking advantage of technology helps to foster reciprocal relationships and deepen that connection with nature. Successively, the children grow a sense of responsibility for the natural world and will be affirmative to protect it. Fortino et al. [99] emphasized the importance of providing direct practical experience to young children to establish this bond and make natural learning meaningful.

Wild environment exposure can reduce children's enjoyment in nature if they feel uncomfortable and frightened. Examples of planned and managed environments for implementing programmes include community gardens, neighbourhood parks, farms, and even zoos. Moreover, school playground programming can improve nature comfort, which provides an additional benefit of a recognised environment that facilitates more successful learning experiences [32]. Educators can brainstorm and model action plans for children that include integrating technological devices with nature learning in the natural environment. These sorts of activities are referenced in Eckhoff's work [63]. Teachers provided children with cameras and iPads to record and document natural surroundings based on their age appropriateness. The outcome is shared in a class with each other's experiences that promote social development and language skills.

## 4. Discussion

In this paper, we have reviewed a range of emergent and divergent technologies that can enhance child–nature interactions and experiences. Generally, within the literature, there is a narrow preconception of technology as an isolating force that distances children from nature [16,24]. However, there is a growing body of the literature that emphasizes the active potential for the curiosity-building and experience-enhancing dimensions of technology too. Both the negative and positive possibilities of emerging technologies need to be considered to better support child–nature connections.

Based on Mahlers' [34,35] child development theory, it is anticipated that young children are frequently expected to go through a separation–individuation process [34] where the growing sense of self in the child separates from the natural world. This psychological individuation is reinforced as a child matures. A sense of nature connectedness, therefore, needs to be nurtured. Further growing digital educational, recreational, and play experiences need to be explicitly linked to nature to provide children with opportunities to help them understand and enjoy connections with nature.

Anderson et al. [15] argued that child-centred interactions are necessary to motivate natural experiences that create meaning and belonging, including tangible or abstract creative learning, expressions, or active collaboration with peers. The ways that technology could be used to improve humans' experiences of the natural environment in their everyday lives must ensure benefits to both nature and the community. The ever-growing sophistication of information technology and the widespread use of mobile phones, Bluetooth, wireless internet, GPS, and other related applications have had a profound effect on children's communication skills. In addition, some experts have suggested that smart digital technologies may reduce the gap between the natural world and 'biophobic' children [7] by ensuring them a familiar, attractive, and playful platform to engage with nature.

Today, digital media plays an important role in young people's activities, as digital media is almost impossible to separate from popular culture and communication styles, especially among children and adolescents [100]. Wills et al. [47] emphasized integrating

digital technology as a bridge to connect children with nature, as children are already familiar with different forms of personal devices. He finds it logical and appropriate to use this opportunity to fill the gap in the exploration of nature and child psychology.

There are several types of research focused on the impacts of screen time on children's development, including potential distraction, gaming addiction, and prioritizing technology over the natural environment [44,48,49,52]. Both positive and negative results can be influenced by things such as how adults utilize and monitor these devices and support children's development and learning [44]. It needs to be recognized that children will become digital citizens in future smart cities; therefore, their freedom to engage in technology-based experiences should not be overlooked but needs to be supported from a young age [37,52]. This explanation is supported by another study conducted by Anderson et al. [15]. Their experience found that visiting parks whilst using mobile applications has no potentially negative consequences, for example, distraction or obliviousness, and does not interfere with children's natural connection and ability to focus on their natural environment.

The last few decades have seen rapid innovations and improvements in the useability of personal digital technologies, and children now access such technologies from a very young age [35] in both households and schools. A frequent mistake is to dismiss the differing experiences of new generations and to expect them to conform to the same patterns of behaviour and standards as previous generations. However, this is not always feasible or advisable, especially when it is not only the technological context that has evolved but also the lifestyle and the way of relating to children that have changed. Over time, technology has become more mobile, accessible, and omnipresent.

Challenges with such technologies remain and are discussed throughout the literature. The current limitations of the technology reviewed in this paper are summarized in Table 4.

**Table 4.** Major technological benefits and limitations addressed in the literature.

| Technology Type | Benefits | Limitations | Reference Numbers |
| --- | --- | --- | --- |
| **Information and Communication Technology through multimedia resources (e.g., computers tablets, interactive whiteboards)** | Multimedia resources, such as videos, images and interactive simulations, provide opportunities to develop children's creativity and communication skills as there is a wider range of information sources available. Enhances teaching effectiveness in school-going children using computer-assisted instruction. Computers and tablets may enable enriching digital play experiences. | Screen time may contribute to sedentary behaviour and decreased physical activity, and reliance on technology may detract from natural exploration. Schools without access to technology may have limited accessibility. Excessive use of technology can negatively impact children's development, but there is a lack of critical synthesis of experiences and play-based pedagogies. | [4,11,13,16,34,35,37,38,43,44,46–51,64,65,69,101–118] |
| **Nature-based application** | Mobile applications and social media promote communication, community building, and healthy digital media use in children as these have additional benefits, such as higher ratings of fun. These can also help children connect with nature and enhance their learning experiences by providing information about local flora and fauna and encouraging outdoor activities. | The use of digital media and technology may have some limitations and risks, such as excessive screen time, less face-to-face communication, distractions, and cyberbullying. There may also be challenges in implementing guidelines and arranging sufficient technology resources for all. | [1–3,12,15,17,19,31,32,39,45,50,54,57,58,60,64–66,70,76,77,98,100] |

**Table 4.** *Cont.*

| Technology Type | Benefits | Limitations | Reference Numbers |
|---|---|---|---|
| **Gamified activities** | In terms of games, mobile applications promote engagement in learning, creativity, problem-solving, social interaction, and teamwork. Location-based games encourage children's interest and knowledge in nature exploration, such as birdwatching, treasure hunt, etc. Some important games have been highlighted in this study: Wobble, Camelot, Geocaching, and Ambient Wood. | The limitations include the need for access to digital devices and an internet connection, gaming addiction, technical difficulties, the potential for distraction and disengagement, increased screen time, and the risk of decreased physical activity. | [5,52,55–57,59,60,62–64,71,72,75,91,96] |
| **Multimodality and extended sensors** | Technology offers children the opportunity to explore natural environments through virtual field trips, 360-degree videos, and virtual reality experiences. These offer the chance to learn about different habitats and species of animals and preserve natural resources. Photography and videography, along with handheld devices, can also help enrich playful experiences in nature. | Risk of negative effects on physical and mental health, social skills, and attention span. | [24,35,36,63,68,74] |
| **Mobile technologies (e.g., digital wireless and wearable devices)** | Technology can also provide children with the appropriate gear and equipment for outdoor activities, such as hiking, camping, or bird-watching. These gears can make outdoor activities more comfortable and enjoyable, ensures safe mobility, and can help children feel more connected to nature. Location-sharing systems allow parents to keep an eye on their children. | Location sharing can impede privacy and natural play behaviour while creating inequities due to the cost of technology. It is challenging to integrate positioning systems seamlessly with children's natural ways of orienting and communicating their location. | [33,42,49,53,59,61] |
| **Mobile augmented reality** | Mobile augmented reality can promote interactive experiences, engagement, and personalized learning while also fostering social interaction and physical activity. Additionally, mobile augmented reality audio systems with binaural microphones could be used to create immersive audio experiences to connect children with nature. | Achieving high precision for location position fix and AR view remains a key obstacle in using augmented reality technology. The use of mobile augmented reality technology may not be accessible to all schools. | [41,54,56,62,71–74,80] |

In summary, there are three major limitations regarding technologies suitable for facilitating and enhancing child–nature relationships. Firstly, suitable technologies are developing rapidly, but such technologies are not evenly distributed within cities or throughout the world. There is an unequal distribution and access to smart technology [101–104]. Secondly, there is a limited agency of children within emerging smart cities. Those conducting and communicating the research can interpret work with children and nature and develop methods that can involve these actors and agents as co-designers of research to bring their viewpoints and specific needs to the foreground [105–108]. Thirdly, the "internet of nature" is an emerging area not well addressed by current research. The generation and prioritising of both a child's eye and nature perspective of technologies are needed to ensure that technology developments genuinely support the experience of children and nature [101,109–112]. Finally, the secondary data was diverse in terms of target groups

(age, gender, socioeconomic condition, and ethnicity), health behaviours and physical environments. There are a lack of studies on the experience of children from different age groups, socio-economic, gendered, and cultural backgrounds [113–118].

The use of technology by young children has increased dramatically. There has been an ongoing debate about this trend and the usage, duration, and guidance of technology use. In many studies, researchers have shown how technological usage, along with other factors, have a positive correlation between nature disconnectedness and children's nature play [35,47,67]. For instance, the question of how children learn through technological games and to what extent should these be allowed in the regular lifestyle. Most early educational experts suggest that the best learning experiences are based on play [11,12,16,17,36,49,61,63,64]. While most practitioners and researchers would agree that familiarity and interaction with technology are crucial for children's future lives, it is difficult to anticipate with any degree of certainty what kinds of digital media will predominate in future homes and workplaces in the next decades. It needs to be understood exactly what nature-based technology is, how to make it equitable, and how it has benefited our society before we can say with certainty that it can mitigate nature deficiency in today's children.

In the process of preparing a tech-savvy child for the future, Plowman and McPake [16] presented a dichotomic point of view among different types of parents. Some parents are eager to make their children independent and help them gain technical ability [50,65,79]. However, not all adopt this approach or take technology use seriously [51]. They question whether their children need to be acquainted with technology at an early age. Some argued that there is no advantage in starting early. Therefore, Fantozzi [48] provides guidelines for the responsible use of digital technology for young children. This approach is particularly relevant for helping children to develop a reciprocal relationship with nature. To achieve this, Fantozzi stresses the importance of intentionality in decision-making around technology use, with appropriate guidance and modeling by both family and teachers. Specifically, the guidelines suggest that tools be placed in the hands of children, with adults working alongside them to ensure their full potential is realised. Equally there needs to be a focus on the process of technology use rather than the end product. By following these guidelines, adults can discern which technologies are developmentally appropriate and can be integrated into children's play in an intentional manner, ultimately benefiting their learning and development [35,48].

It is a society's obligation to build a better future for children. Adults need to work together to promote child rights, health, and agency in a rapidly changing technological landscape. These priorities are not always aligned, and so appropriate models are needed to work through child–nature–technology challenges. Strategies include utilizing technology that promotes active engagement [48]. A paradigm shift in approaches to technology and nature is necessary when it comes to children.

## 5. Conclusions

Presently, there is considerable uncertainty about how digital technologies should be integrated into the nature–technology–children context. This research provides insight into how digital technologies may encourage children to have greater empathy, interest, and engagement with nature. The character and quality of play are rapidly changing in the digital era [65], as technology is becoming a common and easily accessible component of everyday life. This is particularly true for children living in urban areas who have less access to natural environments and greater access to sophisticated technologies.

This review has developed insights into how we might conceptualise nature-sensitive technologies. Questions remain about their impact on how we live and work in cities, and how we integrate nature-based technology into a natural urban setting. It is difficult for urban planners, landscape architects and urban designers to create livable, attractive, safe, and sustainable smart cities when digital infrastructure is constantly evolving and children are learning to communicate in new ways [119,120]. There is a need to understand how technology can act as a resource to support and facilitate nature-based play and

that technology and nature play should not be dichotomized. When implementing and considering digital interventions, it is important to ensure that the focus remains on children and nature rather than on the tools of virtual enhancement themselves. The challenge for urban designers, landscape architects, and planners is to consider how these technologies might interface and connect with urban nature so that they do not hinder or distract from natural settings but rather enhance them. The relationship between nature and technology does not have to be binary; rather, it needs to be reconceptualized as a resource to promote and support the child–nature connection through education, play, and exploration. The planner and architect can include the fundamentals of child-friendly urban design principles while evaluating specific technological extensions.

**Author Contributions:** Conceptualization, R.S. and S.H.; methodology, R.S. and S.H.; software, R.S. and S.H. validation, S.H.; formal analysis, R.S. and S.H.; investigation, R.S. and S.H.; resources, R.S. and S.H.; data curation, R.S.; writing—original draft preparation, R.S.; writing—review and editing, R.S. and S.H.; supervision, S.H. All authors have read and agreed to the published version of the manuscript.

**Funding:** This research received no external funding.

**Institutional Review Board Statement:** Not applicable.

**Informed Consent Statement:** Not applicable.

**Data Availability Statement:** Not applicable.

**Conflicts of Interest:** The authors declare no conflict of interest.

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
