# Peer review of "Reconciling Nature-Technology-Child Connections: Smart Cities and the Necessity of a New Paradigm of Nature-Sensitive Technologies for Today’s Children"

_sustainability, doi:10.3390/su15086453_

Round 1

Reviewer 1 Report

The following advice is offered for the authors to consider areas for improvement to this article:

Thorough proof read required several easily removed mistakes identified.

Table 2 – consistency of presentation – some statements have full stops at the end; remove the random punctuation marks. In addition, Section 2.2 at the end indicates websites identified in a category of “Supporting information” but it is not clear in Table 2 how these were selected/rejected - please consider adding the inclusion/exclusion criteria for the supporting information from websites. There are 64 articles identified for data extraction and analysis, how many websites were included for the purposes of data extraction and analysis?

In Table 2 an article not written in English is classified as ‘unauthentic’; the authors may consider finding a word that does not diminish the perception of value of a research article because it is not in English. Perhaps something more suitable given a binary option is required could be something such as Globally inclusive (English language); Native language only (not available in English).

There is considerable description of various software and smart technologies do but the emphasise of arguments would be better focused on the efficacy of smart technologies/software throughout the article in addressing the research question. For example, where software package properties are described be more critical in identifying evidence of effectiveness avoiding abstract terms such as 'increased understanding' in deductions so that a stronger evidence-based argument is presented that is critical in examining all the issues. What papers and articles present limitations to the technologies? Consider adding an exploration of the limitations from the existing research. 

Author Response

Dear Reviewer 1
Many thanks for your time in organising and completing the review of our paper. We also appreciate the time you spent recommending improvements for our paper. We are pleased to receive a range of positive and constructive comments from the editor and two reviewers.
Reviewer 1 has described a range of recommendations. To address these we have revised the manuscript thoroughly and modified the entire section 2.2. We have provided further explanation of this in the text. The reviewer also suggested communicating the limitations of the study. Limitations have been described within the discussion and conclusion as per reviewer 1’s suggestion.
We have detailed the above-mentioned revisions in the Rejoinder Table below. We thank Reviewer 1 for their help in strengthening the paper on the topic of ‘Reconciling Nature-Technology-Child Connections: smart cities and the necessity of a new paradigm of nature-sensitive technologies for today’s children’. We suggest this revision has improved the clarity of the paper and also better communicated its novelty and innovation in line with the observations of reviewer 1. We argue that our study advances this important topic and puts thoughtful discussion to the readers.
Yours sincerely,
The Authors

Reviewer 2 Report

The topic of this paper is an interesting one that is deserving of attention. The literature review is helpful and highlights the ways that technology  can support children's exploration of nature. 

In some cases the conclusions extend beyond the scope of the actual research (e.g. urban design per se isn't really a focus of the research). Also, I flagged a number of places where the meaning of sentences is unclear - this can be addressed with a detailed review for clarity and grammar (in some cases, the grammar is ok but the meaning of the sentence is very hard to discern). 

Author Response

Dear Reviewer 2
Many thanks for your time in organising and completing the review of our paper. We also appreciate the time you spent recommending improvements for the paper. We are pleased to receive a range of positive and constructive comments from the editor and two reviewers.
Reviewer #2 has described a range of recommendations. To address these, we have revised the manuscript thoroughly and modified it. She/he asked the question “There are 64 articles identified for data extraction and analysis, how many websites were included for the purposes of data extraction and analysis?” We have provided further explanation of this in the text. The reviewer also suggested communicating the limitations of the study.
Reviewer# 2 suggested that the topic of this paper is an interesting one that is deserving of attention. The literature review is helpful and highlights the ways that technology can support children's exploration of nature. We thank Reviewer#2 for appreciating our work. This reviewer also made some important observations in the Conclusion section, which we modified and thank him/ her for pointing out the irrelevancy. Also, they flagged a number of places where the meaning of sentences is unclear - we have modified the necessary sentences and added further explanations where this is useful.
We have detailed the above-mentioned revisions in the Rejoinder Table below. We thank the reviewer for their help in strengthening the paper on the topic of ‘Reconciling Nature-Technology-Child Connections: smart cities and the necessity of a new paradigm of nature-sensitive technologies for today’s children.’ We suggest this revision has improved the clarity of the paper and better communicated its novelty and innovation in line with the observations of Reviewer 2. We argue that our study advances this important topic and puts thoughtful discussion to the readers.

Yours sincerely,
The Authors

Round 2

Reviewer 1 Report

Congratulations to the authors on the considerable effort in achieving the changes in the very short time available. The changes made have produced a much stronger, comprehensive and persuasive set of arguments around the research undertaken.